# Integration of Mesoporous Bioactive Glass Nanoparticles and Curcumin into PHBV Microspheres as Biocompatible Composite for Drug Delivery Applications

**DOI:** 10.3390/molecules26113177

**Published:** 2021-05-26

**Authors:** Arturo E. Aguilar-Rabiela, Aldo Leal-Egaña, Qaisar Nawaz, Aldo R. Boccaccini

**Affiliations:** 1Institute of Biomaterials, Department of Materials Science and Engineering, University of Erlangen-Nuremberg, Cauerstrasse 6, 91058 Erlangen, Germany; aldoleal@yahoo.com (A.L.-E.); qaisar.nawaz@fau.de (Q.N.); 2Escuela de Ingeniería y Ciencias, Tecnológico de Monterrey, Campus Estado de México, Carretera Lago de Guadalupe Km 3.5, Margarita Maza de Juárez, Atizapán de Zaragoza 52926, Estado de México, Mexico

**Keywords:** bioactive glass nanoparticles, composite microspheres, drug delivery systems, PHBV

## Abstract

Bioactive glasses (BGs) are being increasingly considered for biomedical applications. One convenient approach to utilize BGs in tissue engineering and drug delivery involves their combination with organic biomaterials in order to form composites with enhanced biocompatibility and biodegradability. In this work, mesoporous bioactive glass nanoparticles (MBGN) have been merged with polyhydroxyalkanoate microspheres with the purpose to develop drug carriers. The composite carriers (microspheres) were loaded with curcumin as a model drug. The toxicity and delivery rate of composite microspheres were tested in vitro, reaching a curcumin loading efficiency of over 90% and an improving of biocompatibility of different concentrations of MBGN due to its administrations through the composite. The composite microspheres were tested in terms of controlled release, biocompatibility and bioactivity. Our results demonstrate that the composite microspheres can be potentially used in biomedicine due to their dual effects: bioactivity (due to the presence of MBGN) and curcumin release capability.

## 1. Introduction

Biomaterials to be used in dental and orthopedic applications should exhibit the ability to form a strong bond with bone tissue and to promote bone growth. Bioactive glasses (BGs) exhibit the formation of hydroxyapatite on their surfaces when in contact with body fluids, which has been considered to induce osteoconductivity and strong bonding to bone tissue [1,2,3,4]. Macro devices based on these biomaterials have been studied since the last decade on different surgical and biomedical applications. However, recently, new processes have been explored to produce some nano structures such as mesoporous bioactive glass nanoparticles (MBGN), nanofibers, among others, in order to enhance BGs bioactivity and to expand the applications in the nanomedicine [5,6]. In this context, the combination of MBGN with different biomaterials at the micro and nanoscale is gaining increasing attention for applications ranging from tissue engineering to drug delivery systems [7,8,9,10]. However, the assimilation of MBGN in biological environments has resulted in being difficult for certain concentrations. Some authors have reported the rapid exchange of ions between the MBGN and the aqueous medium, which may result in a rapid and undesired increase of the pH of the medium [11,12]; this could make the biocompatibility and assimilation of MBGN in biological environments difficult. Another possible cause of these effects could be due to the interaction presented between nanoparticles and cells, in which the contact with the cell membrane is different than with devices of bigger sizes; this can also be considered an important cause of the side effects observed. Then, a suitable approach to damping these possible effects is the combination of MBGN, and BGs, in general, with other biomaterials [7,8,9,10]. Specifically, composite structures such as films and coatings for 3D scaffolds have been developed in the last few years [10,13,14,15]. 

Due to their high biocompatibility and biodegradation, polyhydroxyalkanoates (PHAs) have been widely utilized in the fabrication of different biomedical devices, such as films, three-dimensional scaffolds, micro and nano particles, and polymer-based drug carriers [16]. Specifically, the use of PHAs as drug delivery systems has been previously explored for the fabrication of micro and nanoparticles for the loading and release of different substances, antibiotics and natural extracts (phytotherapeutics) [17,18,19,20,21,22]. In this field, a particular phytoherapeutic, curcumin, has exhibited a series of attractive properties such as antioxidant and anti-inflammatory effects in addition to being favorable for wound healing [23,24]. There are some reports about the successful entrapment and release of curcumin by micro and nanoparticles made of various materials, PHAs, among others [22,25,26]. 

In this work, poly(3-hydroxybutyrate-co-3-hydroxyvalerate) (PHBV), a member of the PHA family, was combined with MBGN at different ratios to form composite microspheres which were loaded with curcumin. The biocompatibility of the composite microspheres was assessed by cell culture studies, and the release rate of curcumin and the bioactivity of the microspheres were investigated.

## 2. Results

### 2.1. Microsphere Size and Zeta Potential

The average diameter of the composite microspheres developed in this work was 2.1 ± 0.3 µm (Figure 1). No significant difference between composite microspheres and blank PHBV microspheres (not containing MBGN) was found. In addition, the size was found to be similar to previous reports on PHA based microspheres fabricated by using similar emulsion techniques [13,22,27,28]. On the other hand, the size of MBGN (Figure 1) was 177 ± 13 nm, also similar to values previously reported in literature [6,29,30].

An important aspect in the use of particles as drug carriers is its capacity to remain homogenous and disperse in liquid environments. One of the properties that impacts this behavior is the Zeta potential of the particles, which can promote the repealing or the attraction of the particles in different environments. The Zeta potential of the composite microspheres was −14.30 ± 1.33 mV (measured in PBS at pH: 7.5), which indicates that the fabricated microspheres should not tend to agglomerate in aqueous environments similar to PBS. Pure PHBV microspheres and MGBN samples were measured exhibiting a zeta potential of −4.58 ± 0.14 mV and −13.13 ± 0.4 mV, respectively. 

### 2.2. Surface Morphology of Composite Microspheres

Homogeneity on the spherical shape of the composite microspheres was observed for the PHVB/MGBN ratios of 90:10, 80:20, and 70:30. A similar morphology was observed in pure PHBV microspheres. Typical micrographs are compared in Figure 2a,b. For these three conditions, the shape of microspheres was not affected by the incorporation of MBGN. However, for higher MBGN ratios (60:40 and 50:50), the morphology was modified and, apparently, as the MBGN ratio increased, it exceeded the capacity of the PHBV to form the microsphere matrix adequately. Examples of this can be observed in Figure 2c in which some microspheres showed MBGN aggregation outside the PHBV matrix and, in Figure 2b, where a portion of the MBGN was even observed dispersed around the microspheres and not fully covered by the PHBV matrix.

### 2.3. Composition Analysis 

Figure 3 shows the FTIR spectrum of the composite microspheres at the 70:30 PHBV/MBGN ratio. The pattern exhibits peaks at 455 cm^−1^ and 1067 cm^−1^, which correspond to Si–O–Si stretching and Si–O–Si bending modes, respectively, present in the MBGN [6]. However, peaks may possibly overlap with the more intense spectra of the PHBV. The peak at 1731 cm^−1^ indicates the stretching mode of C=O in the crystalline phase of PHBV. In addition, the peaks at 1282 cm^−1^ and 1179 cm^−1^ correspond to the crystalline and the amorphous parts via the stretching modes of C–O–C present in PHBV. On the other hand, the peaks at 817 cm^−1^ and the peak at 3509 cm^−1^ correspond to the vibrations of -CH bonds and the O-H stretch, respectively, both characteristic in curcumin. As in the MBGN case, these peaks may be overlapped by the spectra of PHBV. However, the band around 3400 cm^−1^, due to the stretching of hydrogen bonded -OH groups present in curcumin, was clearly observed [31,32,33]. Due to the fact that no new functional groups in the FTIR spectrum were detected, it is possible to consider that the composite is simply a blend of the different components without the formation of any other intermediate phase. This result is similar to previous reports using the same materials [13,34].

To support the composition of the composite, the XRD diffractogram of composite microspheres at 70:30 PHBV/MBGN ratio is shown in Figure 4. The characteristic peak of crystalline curcumin present at 2Ɵ 26.5° can be observed [35]. However, the peak for curcumin at 14.3° and the amorphous structure of MBGN at 2Ɵ 23.1° may also be overlapped by the more intense PHBV spectrum, from which several characteristic peaks at 2Ɵ 13.4°, 16.9°, 21.3°, 22.4°, 26° and 27° are clearly observed [6,35]. Similar results have been reported in previous studies involving some of these materials [13,34].

### 2.4. Curcumin Entrapment Efficiency

Figure 5 shows the curcumin entrapment efficiency (CEE) of the different microspheres fabricated. It was observed that the incorporation of MBGN at the different ratios studied in this work apparently did not affect the entrapment of curcumin at the PHBV/curcumin ratio of 90:10. The CEE was similar and around 90% for all the conditions studied. This indicates that, as expected, curcumin can be entrapped by the PHBV/MBGN composite and processed through emulsion techniques, at least under the MBGN ratios studied in this work.

### 2.5. Curcumin Release Kinetics in PBS

One of the advantages of using PHA-based micro particles as a drug carrier is the possibility of controlling and prolonging the drug release [17,22]. Figure 6 shows the curcumin release kinetics of the 90:10 (PHBV/MBGN) sample in PBS. The curve exhibits a typical release behavior similar to that reported in previous studies [17,22,36,37].

### 2.6. In Vitro Cell Culture Assays

As previously mentioned, one aim of this work is to reduce possible side effects due to different concentrations of MBGN in biological environments, by incorporating MBGN into a PHBV matrix and by adding curcumin. The comparison of the cell viability, at two different time points, between the composite microspheres and free MBGN was used to determine a possible side effect due to the high reactivity of MBGN. To this purpose, we compared the cell viability, after 24 h and 7 days of incubation, respectively (Figure 7 and Figure 8). The free MBGN condition was calculated to contain similar concentrations of MBGN to that in the highest ratio in the composite microspheres (50:50). 

In Figure 7, it is possible to observe a similar cell viability between the experiments carried out with 90:10 and 80:20 MBGN ratios and the control. This was more evident in 10^−2^ and 10^−3^ dilutions. However, in samples with the highest MBGN ratio (50:50), only in 10^−2^ and 10^−3^ dilutions was the cell viability similar to the control in the first 24 h. In contrast, samples treated with free MBGN exhibited a cell viability considerably lower compared to the control. This was also observed between the 10^−1^ dilution of the highest ratio of MBGN (50:50) condition and the control. 

In addition, fluorescent microscopy observations were used to analyze the proliferation and morphology of the cells after exposure to the composite microspheres and MBGN. Blue staining represents the cell nuclei and red staining represents the cell cytoplasm. Figure 8 shows the fluorescent micrographs of cells treated with the highest dilution of microspheres at 50:50 PHBV/MBGN ratio (Figure 8a) and cells treated with the same dilution of free MBGN (control) (Figure 8b), both after 24 h of incubation. In Figure 8a, the sample treated with composite microspheres, most of the cells exhibited a classical reported morphology (colonies conformed by homogeneous flat and spread cells). On the other hand, Figure 8b shows that cells have different morphology (smaller and less spread cells, with heterogeneous morphology and apparent low cell–matrix attachment surface, compared to control analysis), possibly due to damage caused by the direct exposure to the free MBGN, and/or changes of pH produced by these scaffolds.

Figure 9 shows the cell viability after seven days of incubation. In this time set, the cell viability of samples treated with free MBGN decreased considerably in comparison to the control. In contrast, most of the samples treated with composite microspheres (90:10, 80:20 and most of the dilution folds at 70:30) exhibited similar cell viability to the control. 

Fluorescent microscopy analysis was developed after this time lapse of exposure to the composite and free MBGN. Figure 10 shows an image of cells treated with a dilution of microspheres at 50:50 PHBV/MBGN ratio, and the corresponding amount of free MBGN dilution after seven days of incubation. 

Figure 10a shows the presence of a higher number of living cells with respect to control experiments, which was determined based on the qualitative analysis of stained nuclei. On the other hand, Figure 10b shows a qualitatively lower population of living cells, based on less presence of nuclei. In addition, seeded cells showed a heterogeneous and irregular morphology, compared to control experiments. These observations can be explained due to the long exposure to a high concentration of free MBGN microspheres. 

### 2.7. Bioactivity Assessment

Finally, bioactivity assays were carried out involving the characterization of the formation of hydroxyapatite (HA) on the surface of the microspheres after immersion in SBF. For this assay, the condition with the lowest proportion of MBGN (90:10) was selected. Figure 11a shows a specific section of composite microsphere diffractograms before (i.e., yellow curve) and after seven days (i.e., blue curve) being immersed in SBF. 

As these figures show, some of the peaks related to the semi crystalline phase of hydroxyapatite appeared at 2Ɵ 29° and 32°, agreeing to similar previous literature reported [6,13,14,34,38]. In the case of the composite microspheres, calcium ions released from the MBGN likely interact with the ions present in the SBF, promoting the formation of hydroxyapatite on the surface of the microspheres. 

The formation of this layer on the surface of the composite microspheres was confirmed by SEM observations after seven days of immersion in SBF (Figure 11b). The morphology is similar to previous observations during the formation of hydroxyapatite on BG-based structures [6,30,38]. Thus, both characterization methods confirmed the bioactivity of the composite microspheres at the lowest MBGN content (90:10) studied in this work.

## 3. Discussion

### 3.1. Microsphere Size and Z-Potential

When using biomaterial-based carriers in drug delivery, the particle size may modify the interaction with cells [22,39]. Thus, depending on the application field, the particle size could exhibit advantages or disadvantages. For example, it has been observed that smaller particles could damage the cell membrane [40], while large particles show restrictions during endocytosis [22]. Due to the size of MBGN (177 ± 13 nm), in addition to the characteristic release of cations, a rapid change on the pH could appear, and these can produce cytotoxic effects in biological environments [11,12]. In contrast, the composite microspheres (2.1 ± 0.3 µm) will interact with the cell membrane in a different way, and such microspheres have a suitable size, which should enable them to enter into blood vessels without causing hemodynamic problems.

For the different MGBN ratios used in the fabrication of the composite microspheres, an apparent saturation limit of MBGN embedded in the PHBV matrix can be observed. Furthermore, as the MBGN ratio increased, the composite particles lost their spherical shape and homogeneity (Figure 2c,d). This can be the result of differences in the material chemical composition, which impacts hydrophobicity and affects its affinity [41,42]. However, most of the PHBV/MBGN ratios studied in this work (90:10, 80:20 and 70:30) exhibited a good integration of the MBGN into well-formed PHBV microspheres. Composition analysis of the composite microspheres supports the non-reactive interaction of MBGN and PHBV similar to previous reports [10,13,14,15]. 

Another desirable characteristic for particulate drug carriers is the possibility to obtain homogeneously dispersed particles in a liquid carrier. Each component of the present microspheres has been studied extensively in previous works, and differences in zeta potential, hydrophobicity and affinity with aqueous media have been reported [41,42,43]. The zeta potential of the composite microspheres indicates that their agglomeration may be hindered in aqueous environments similar to blood. These characteristics ensure the uniformity of properties and stability of the composite microspheres to be applied as drug carriers in biological environments.

### 3.2. Composite Microspheres as Drug Delivery Systems

For its application as drug delivery system, it is important that the carrier is physiochemically compatible with the drug to be loaded. A model drug is commonly defined as the component that can be measured/traced once released from the device [22]. The compatibility depends on many factors, such as the hydrophobicity of drug and carrier, the solubility of the drug in the solvent used for the carrier fabrication, the stability of the drug during the fabrication process, among others [22]. In this work, curcumin was selected as a model drug due to its beneficial properties, useful in this application, and its compatibility with the organic solvent used in the fabrication of the microspheres [23,24]. Curcumin has been successfully entrapped in different biomaterials with controlled release purposes [19,25,26]. In contrast to other drugs, such as gentamicin, which has exhibited low compatibility with PHAs in the past (i.e., having an entrapment efficiency below to 50%), curcumin has been reported to have good compatibility and entrapment efficiency in PHA matrices [18,19,22]. 

Tailoring drug release behavior is important since a controlled drug release can allow prolonged treatments with reduced administration of the drug, with the purpose of decreasing potential side effects due to the presence of high drug concentrations in plasma during the early stages of treatment. The curcumin release kinetic of composite microspheres (Figure 6) exhibited a biphasic behavior in which the first phase, the so-called “burst” phase, appeared during the first 5 h approximately. The appearance of this phase can be explained, as previously reported [17,22], due to the interaction of the medium with the surface of microspheres and the relation between volume, size and surface area. As it has been reported, the rate of release is influenced by the size of the particles [18,22]. 

The next phase, the “controlled release phase”, appeared after approximately 5 h and continued until 20 h in the present experiment. After 25 h, measurements at random periods of time were taken with no significant changes in concentrations observed. This release behavior was similar to previously reported results using PHA based microspheres as drug release systems [17,22,36,37]. The CEE (over 90% in all cases) and the release kinetic curve showed the potential of the composite microspheres for achieving prolonged release, as a delay of the maximum curcumin concentration was observed during the first 10 h. The release kinetics depends on many factors, such as the morphology of the microspheres and the drug solubility in the environment, among others. Then, the release kinetics may be modified by varying these factors and the possibility for entrapping other pythotherapeutics and even to combine inorganic/hydrophilic components with organic/hydrophobic components through the same matrix could be explored in further studies.

### 3.3. Cytocompatibility of Composite Microspheres

The adequate incorporation of MBGN into the PHBV matrix modifies the interaction between the composite and the cells in most of the ratios studied. The PHBV matrix serves as an interface which moderates the release and the direct contact of MBGN with the cell culture medium. This effect could avoid high initial MBGN concentrations in the biological environment and, due to the size of the microspheres, it provides a contact interaction which reduces the possible side effects of MBGN directly exposed to cells [11,12,18,40]. This is supported by the trend of cell viability observed in the WST-8 results through the different ratios studied in this work. Additional information is provided for the first time by the analysis of the fluorescent microscopy images, in which the samples treated with composite microspheres (Figure 8a and Figure 10a) exhibited a larger group of healthy cells, in terms of the number of nuclei observed, and the cell morphology and proliferation patterns presented in these samples, indicating a more viable interaction for the cells due to the composite. This in contrast to the observations in cell samples treated with similar amounts of free MBGN (Figure 8b and Figure 10b), which showed abnormal patterns of cell proliferation and morphology. 

Although at higher MGBN concentrations the cell viability was lower compared to the control, or even in the presence of microspheres containing lower MBGN concentrations (i.e., 90:10 and 80:20), significant differences between the living cell population could be observed by the WST-8 viability results. This, in combination with the fluorescent microscopy observations, brings evidence of a clear different behavior in vitro, between free MBGN and similar amounts of MBGN but administrated through microspheres. There is still ongoing discussion about what phenomena, the pH change due to the rapid release of ions or the effect of size of the nanoparticles in the interaction with cells, have a bigger impact on the reduction of the cell viability due to free MBGN.

Additionally, even though a study to measure the effect of curcumin on cells was not performed in this work, the slight difference observed between the WST-8 cell viability on samples treated with only PHBV and the composite microspheres (Figure 7 and Figure 9), showed that the beneficial properties of curcumin [23,24] may play a role in the damping of possible side effects due to the presence of (high concentrations of) MBGN. The morphology and proliferation behavior observed in cells treated with composite microspheres could also be explained due to the release of curcumin. In almost all the conditions studied for the composite microspheres, the cell viability was higher than in its respective dilution of free MBGN. Finally, the formation of a layer of hydroxyapatite on the surface of microspheres, even for the condition with the lowest MBGN ratio (90:10), after contact with SBF, confirmed the bioactivity of the composites microspheres, which is the result of the exchange of ions from the composite to the medium. A little difference observed between the surface morphology of the hydroxyapatite layer formed on the microspheres in comparison to the layer formed on MBGN after similar time lapses may indicate an initial stage of hydroxyapatite layer formation in the case of the composite microspheres. This could be explained due to the interaction with NBGN, and SBF is delayed by the PHBV polymer matrix. In correspondence to this, composite samples with higher MBGN ratios will exhibit a higher ion exchange and less delay on the formation of the hydroxyapatite layer, indicating the potential of the composite microspheres to induce local mineralization.

## 4. Materials and Methods

### 4.1. Materials

PHBV was purchased from Goodfellow (Bad Nauheim, Germany), dichloromethane from Sigma-Aldrich (St. Louis, MO, USA), curcumin from Sigma-Aldrich (Steinheim, Germany) and Poly Vinyl Alcohol (PVA) from Baxter Healthcare (Opfikon, Switzerland). Tetraethyl orthosilicate [TEOS]–99%, triethyl phosphate [TEP]–99% and calcium nitrate were bought from Aldrich (Steinheim, Germany). Furthermore, ethyl acetate, cetyl-trimethylammonium bromide [CTAB] were provided from Merck (Darmstadt, Germany), ammonium hydroxide 28% from VWR (Fontenay Sous Bois, France), distilled water (MilliQ), and absolute ethanol–99.8% from Alfa Aesar (Kandel, Germany). All chemicals used were of an analytical grade.

### 4.2. Synthesis of MBGN

The mesoporous bioactive glass nanoparticles were produced by using a modified Stöber procedure as reported by Nawaz et al. [6]. Briefly, 0.56 g of CTAB was dissolved in 26 mL of water and stirred for 15 min. Then, 8 mL of ethyl acetate was dropped carefully into the CTAB solution. Afterwards, 3 mL of TEOS were added under continuous stirring; then, ammonium hydroxide was added to maintain the pH at 10.5. Later, calcium nitrate and TEP were added carefully to the solution and then allowed for reacting under stirring for 3 h. Finally, the suspension was centrifugated at 7000 rpm in a 5430R from Eppendorf (Hamburg, Germany) for 10 min and washed with ultrapure water three times. The precipitates were dried in an oven at 60 °C overnight and then calcinated at 700 °C with a heating rate of 2 °C/min for 5 h.

### 4.3. Composite Microsphere Fabrication

For the fabrication of microspheres, a modified solid in oil in water emulsion (S/O/W) method was used, similarly to the technique reported in previous works [18,22]. Different amounts of MBGN were added into the PHBV solution in 10 mL dichloromethane. In this work, the concentrations of MGBN in the PHBV microspheres started at 10% (*w*/*w*) and increased gradually by 10% until reaching 50% (*w*/*w*) to obtain five PHBV/MBGN ratios of 90:10, 80:20, 70:30, 60:40 and 50:50 (*w*/*w*). Then, curcumin was loaded in all conditions at 90:10 (*w*/*w*) (PHBV/curcumin) ratio and mixed at 800 rpm to form the S/O phase. An aqueous solution of 1 mg/mL of PVA was prepared and mixed at 600 rpm to conform the W phase. The resulting solution was mixed at 19,000 rpm using a homogenizer T18 (IKA, Staufen, Germany). Afterwards, the emulsion was centrifuged and washed with ultrapure water twice, and the supernatant was completely removed. The microspheres precipitated were dried in an incubator at 60° for overnight and then stored protected from the light.

### 4.4. Curcumin Entrapment Efficiency

The Curcumin Entrapment Efficiency (CEE) was determined according to the supernatant method previously reported [22,44]. Briefly, microsphere samples were immersed in ethanol and then supernatant curcumin concentration was measured through a UV/Vis spectrophotometer Specord 250 (Analytikjena, Jena, Germany) at 425 nm. The entrapped efficiency was calculated by using the following equation: (1)$$CEE=\frac{(M_{the}-M_{sup})\left(100\%\right)}{M_{the}}$$
where $M_{sup}$ is the mass of curcumin measured in the supernatant (mg), and $M_{the}$ is the theoretical initial mass of curcumin added during the fabrication of the samples (mg).

### 4.5. Microsphere Surface Morphology

The surface morphology of the composite microspheres was observed by Scanning Electron Microscopy (SEM) using a microscope Auriga (Zeiss, Munich, Germany). The microspheres were sputter coated with gold in a turbomolecular pumped coater Q150T Plus (Quorum, Laughton, UK), prior to SEM examination.

### 4.6. Microspheres Size and Zeta Potential Analysis

The size average measurements of the composite microspheres, MBGN and blank PHBV particles were performed by using a Zetasizer NanoZS (Marlvern, Worcestershire, UK) in deionized water. The zeta potential measurements of composite microspheres were carried out by suspending samples in PBS at pH: 7.5, MGBN and blank PHBV in deionized water.

### 4.7. Structural Characterization of Composite Microspheres

The structural characterization of the composite microspheres was determined through FTIR by using a Nicolet 6700 Thermo Scientific FTIR spectrometer (Waltham, MA, USA). Measurements were carried out in absorbance mode at wavenumber range from 4000 to 400 cm^−1^ at a resolution of 4 cm^−1^. X-ray diffraction (XRD) was performed by using an X-ray diffractometer D8 advance Brucker (Billerica, MA, USA) in the range of 10° to 80° 2Ɵ with a step size of 0.01° and dwell time of 1° per minute.

### 4.8. Bioactivity Assessment

Samples of microspheres at a 10:1 ratio of PHBV/MBGN were immersed in Simulated Body Fluid (SBF) for 7 days. SBF was produced according to the protocol of Kokubo [45]. Then, SBF was removed, and microsphere samples were stored and dried at room temperature for examination. The surface morphology of samples was analyzed as in Section 4.5. The XRD analysis of microsphere were performed by using X-ray diffractometer D8 advance Brucker (Billerica, MA, USA) in the range of 2Ɵ 26° to 39° with a step size of 0.01° and dwell time of 1° per minute. 

### 4.9. Curcumin Release Kinetics

For the curcumin release kinetics, 10 mg of composite microspheres were placed in sterile falcon tubes with caps by triplicates. Then, 10 mL of Phosphate Buffer Solution (PBS) at pH = 7.4 (Sigma-Aldrich, Steinheim, Germany) was added into each falcon tube and stored at room temperature. The release kinetic curve was obtained in accordance with previously reported methods [19,22] by using the spectrophotometer Specord 250 (Analytikjena, Jena, Germany) at 425 nm. 

### 4.10. Cell Culture Assays

Human osteoblasts like MG-63 cells (sourced from the Biomaterials Institute cell bank) were grown in Dulbecco’s Modified Eagle Medium (DMEM) supplemented with 10% FBS (Gibco, Dreieich, Germany). In addition, 50,000 cells were seeded on 24 well-plates and incubated for 12 h before performing the biocompatibility tests.

Master solutions of 1.0 mg/mL of composite microspheres (from each PHBV/MBGN ratios), PHBV microspheres without loading and free MBGN were prepared. Afterwards, ten-fold serial dilutions (10, 100 and 1000 µL) from each master solution were added to the 24 plates already containing seeded cells. Experiments were carried out in triplicate. The cell viability was assessed after 24 h and 7 days post treatment by WST-8 assays (Sigma-Aldrich, Steinheim, Germany) according to the provider instructions. Briefly, the well-plates’ supernatant was removed and washed with PBS; then, a freshly prepared cell culture medium was added containing 1.0% *v*/*v* WST-8 solution, followed by incubation for 2 h. Subsequently, 100 µl of supernatant from each well-plate was transferred into a 96 well-plate for absorbance measurement at 450 nm in a PHOmo Autobio (Labtec Instruments, Zhengzhou, China).

### 4.11. Cell Staining

Samples were treated with Fluorescent DNA stain (DAPI) and Vybrant DyeCycle staining (both reagents from Merck, Darmstadt, Germany) for observation in a microscope AxioCam ERc 5s Primovert (Zeiss, Munich, Germany), after 24 h and after 7 days of incubation. Images were analyzed by using the open-source software Fiji (ImageJ, Bethesda, MD, USA).

### 4.12. Statistical Analysis

Data were presented as the mean ± standard deviation of each treatment. Data were analyzed for statistical significance using the analysis of variance One-Way ANOVA followed by Tukey test (*p* < 0.05), using software Origin 8 (OriginLab, Northampton, MA, USA).

## 5. Conclusions

In this work, the combination of MBGN, curcumin and PHBV as a composite drug delivery system was successfully investigated. The composite microspheres exhibited an average diameter of 2.10 ± 0.27 µm, spherical uniform shape and a semi-homogeneous distribution under certain MBGN concentrations. The zeta potential values in PBS indicated that composite microspheres should not tend to agglomerate in aqueous environments with pH similar to blood. The curcumin entrapment efficiency was around 90%, and it was not affected by the incorporation of MBGN in the ratios studied. The curcumin release kinetics showed signs of a gradual release, and the presence of bioactivity was confirmed by immersion tests in SBF. The in vitro results exhibited a significant difference between the biocompatibility of samples treated with the composite microspheres compared to the ones treated with free MBGN. This indicates a reduction of the possible side effects, due to rapid alkalinization of the medium by MBGN and effects of the interaction between cells and nanoparticles, when they are administered through the composite microspheres. The cell compatibility was measured at two different time points and was supported by WST-8 additional evidence about the behavior of cells due to the presence of particles being developed by fluorescent microscopy. The results confirmed a different cell viability when comparing the administration of similar amounts of free MBGN and composite microspheres. Fluorescent observations provided information for the first time about the behavior of composite microspheres and high ratios of free MBGN in contact with the cells. The potential of the composite microspheres as a drug delivery system was also confirmed. The present composite microspheres are in principle suitable for applications in which both curcumin drug release and bioactivity are required, such as in bone surgical procedures or in the development of bone regeneration therapies where curcumin and the presence of hydroxyapatite can act to promote bone healing and bone tissue formation. However, long-term cell biology studies are required for the further development of these devices and to consider their applications in bone regeneration approaches.

## Figures and Tables

**Figure 1 molecules-26-03177-f001:**
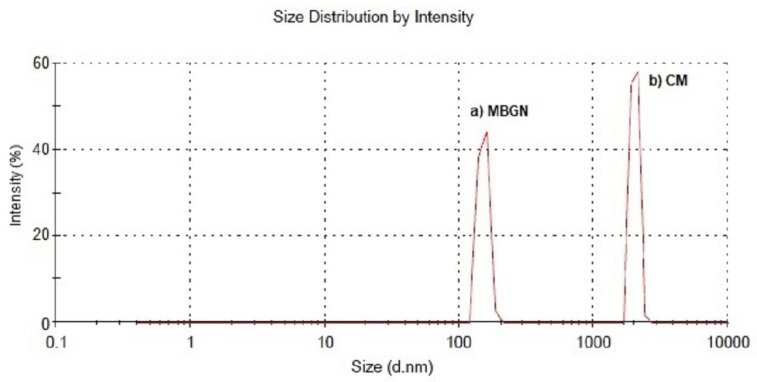
Average particle size distribution of (**a**) MBGN and (**b**) composite microspheres (CM) measured by laser diffraction.

**Figure 2 molecules-26-03177-f002:**
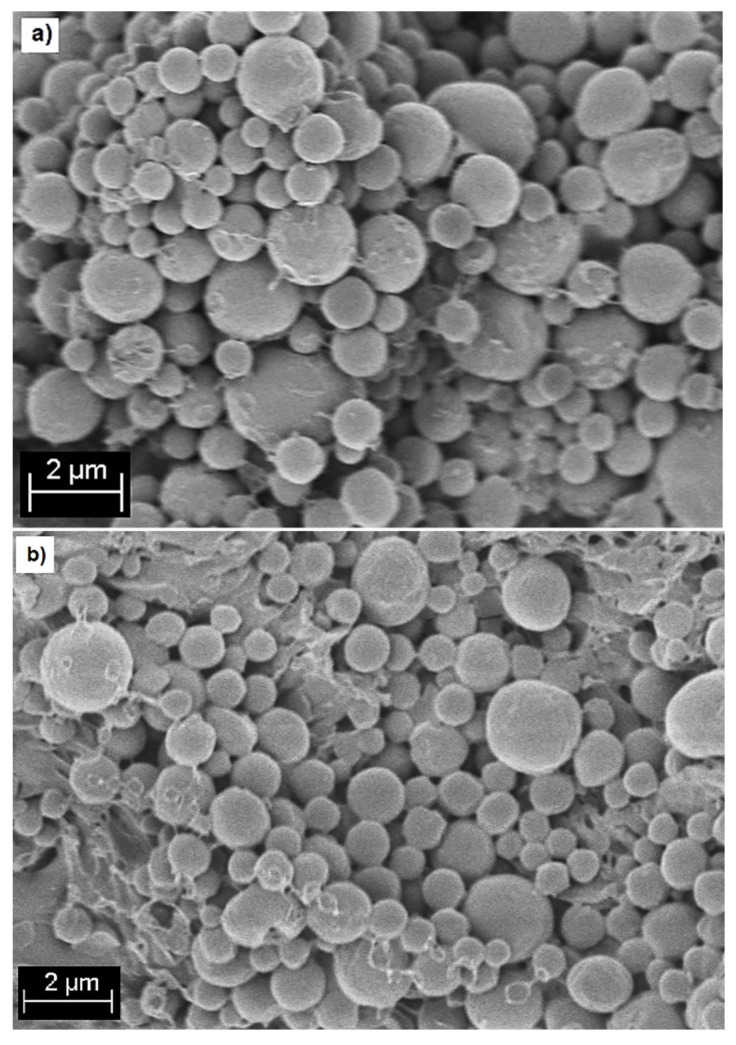
SEM micrographs of (**a**) blank PHBV microspheres; (**b**) PHBV/MBGN composite microspheres at 90:10 ratio, and PHBV/MBGN composite microsphere at higher ratios; (**c**) 60:40; and (**d**) 50:50.

**Figure 3 molecules-26-03177-f003:**
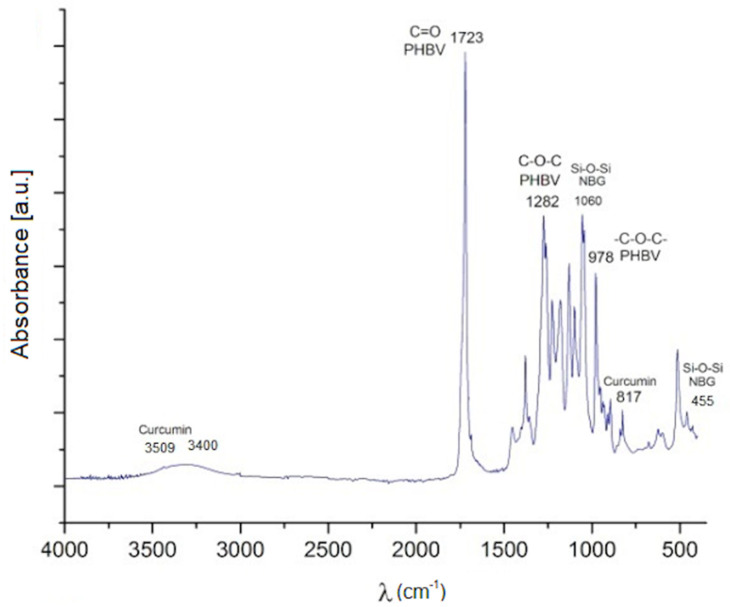
FTIR spectrum of a composite microsphere (70:30 PHBV/MBGN ratio). The detected peaks are discussed in the text.

**Figure 4 molecules-26-03177-f004:**
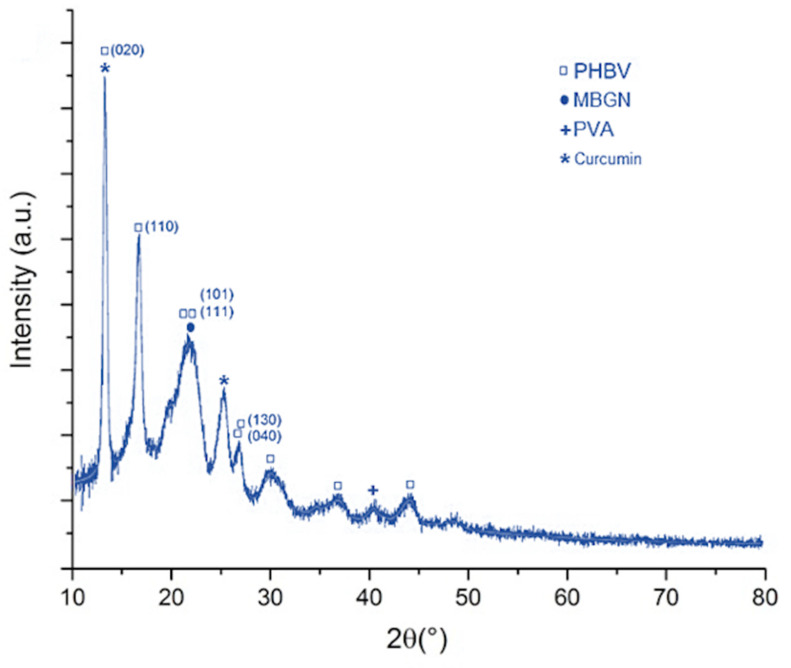
XRD pattern of a composite microsphere (70:30 PHBV/MBGN ratio). The identified peaks are discussed in the text.

**Figure 5 molecules-26-03177-f005:**
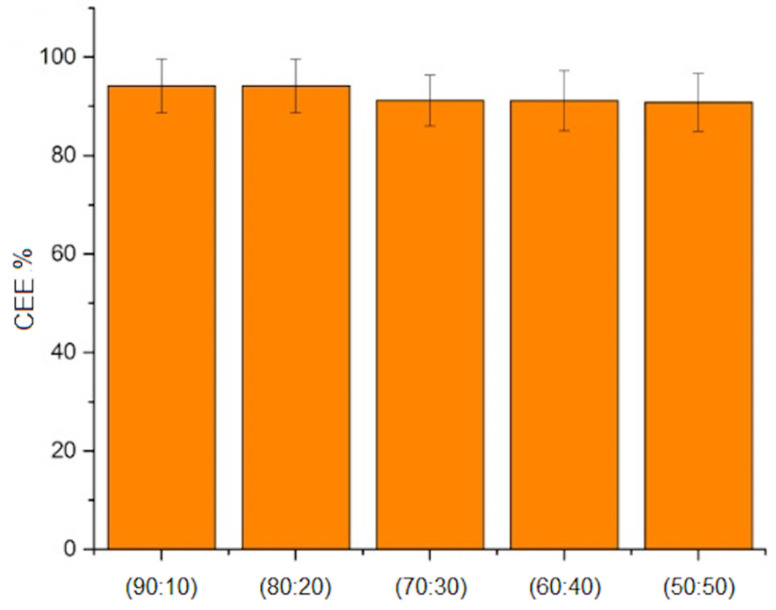
Curcumin entrapment efficiency (CEE) in microspheres at different PHBV/MBGN ratios (error bars show the Standard Deviation).

**Figure 6 molecules-26-03177-f006:**
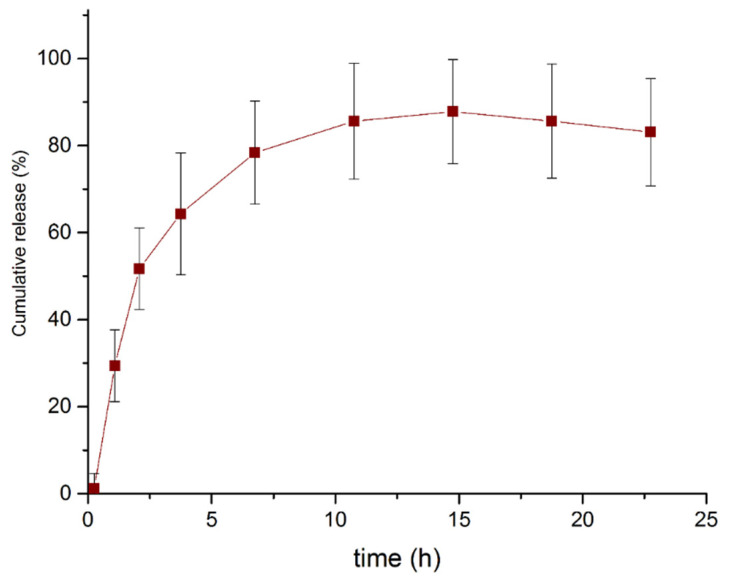
Cumulative release kinetic of curcumin from composite microspheres in PBS at a 90:10 (PHBV/MBGN) ratio in PBS.

**Figure 7 molecules-26-03177-f007:**
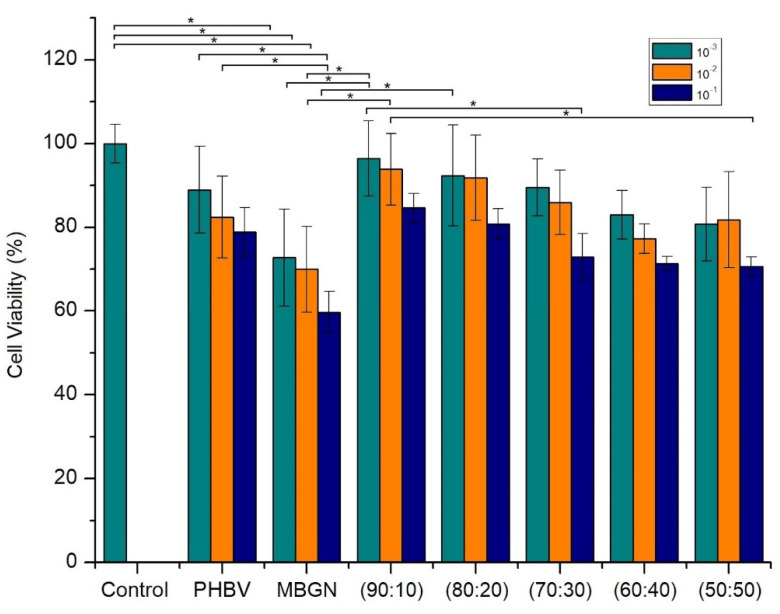
Cell viability of MG-63 cells treated with control (cells without any treatment), pure PHBV microspheres and composite microspheres at different MBGN ratios (ten-fold dilutions of 1 mg/mL), after 24 h of incubation. (* corresponds to *p*-value ≤ 0.05).

**Figure 8 molecules-26-03177-f008:**
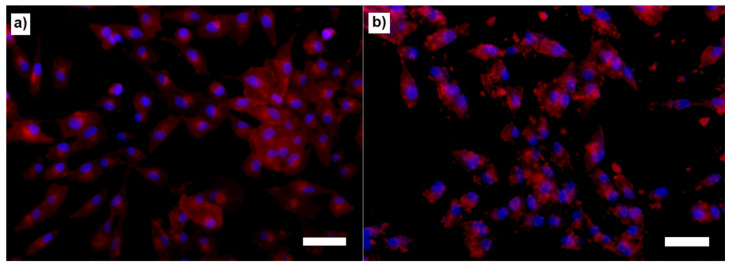
Fluorescence micrographs of MG-63 cells treated with (**a**) composite microspheres at 50:50 (PHBV/MBGN) ratio and a (**b**) similar amount of free MBGN after 24 h (bar size is 10 µm).

**Figure 9 molecules-26-03177-f009:**
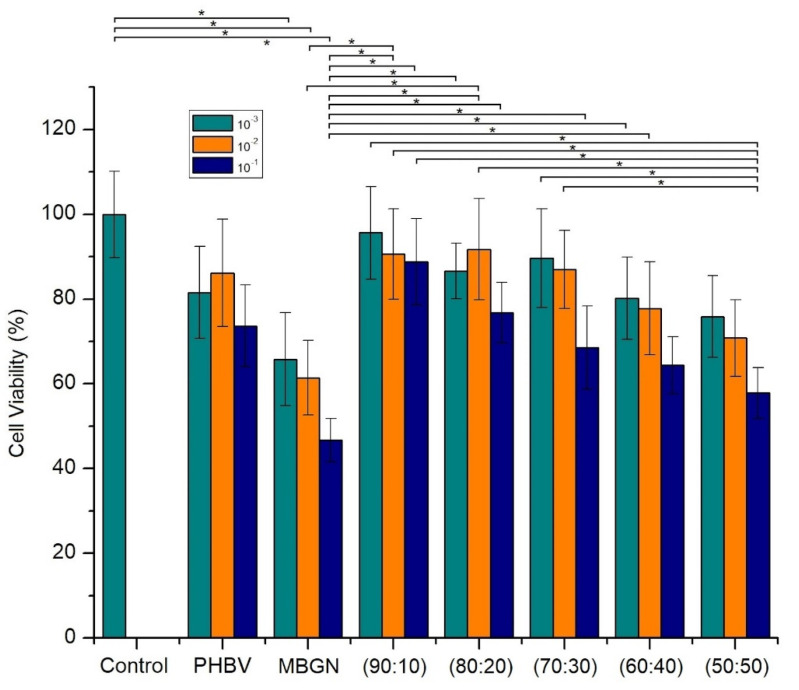
Cell viability of MG-63 cells treated with control (cells without any treatment), pure PHBV microspheres, and microspheres at different MBGN ratios (ten-fold dilutions of 1 mg/mL), after seven days of incubation. (* corresponds to *p*-value ≤ 0.05).

**Figure 10 molecules-26-03177-f010:**
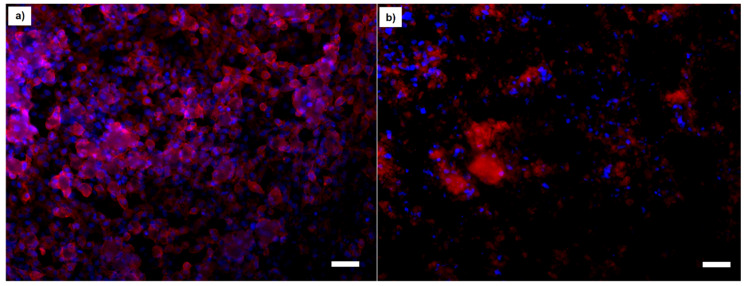
Fluorescence microscopy micrographs of MG-63 cells treated with (**a**) composite microspheres at 50:50 (PHBV/MBGN) ratio and (**b**) same proportion of free MBGN, after seven days of incubation (bar size is 20 mm).

**Figure 11 molecules-26-03177-f011:**
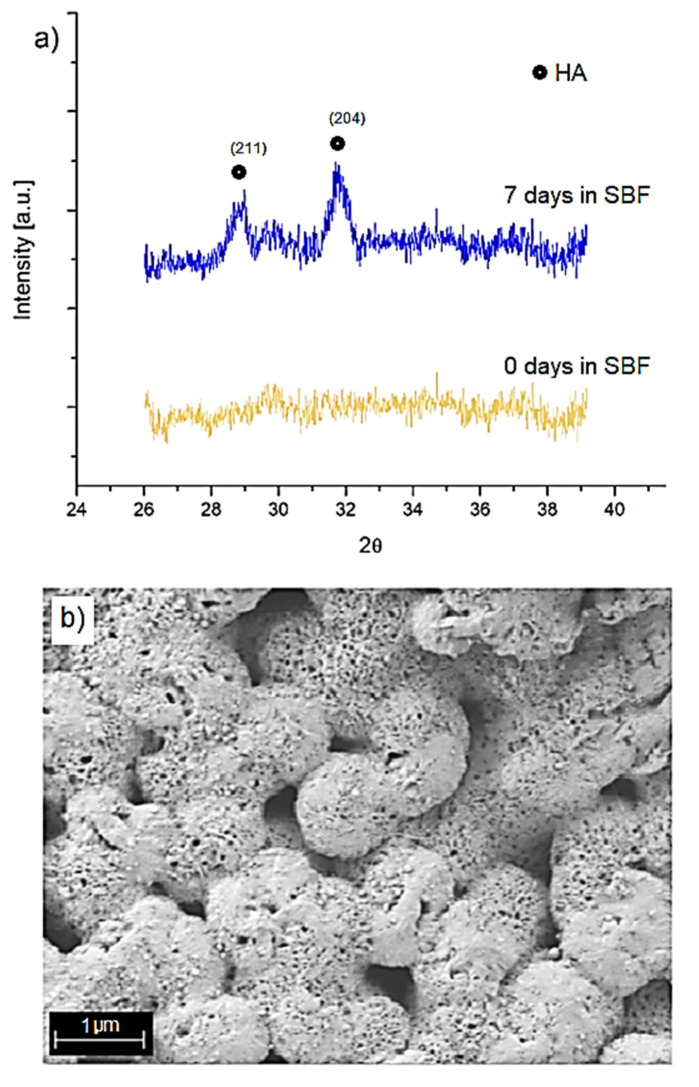
(**a**) XRD patterns of composite microspheres PHBV/MBGN (90:10) after zero and seven days of immersion in SBF, and (**b**) SEM micrograph of composite microspheres (same ratio) after seven days of immersion in SBF.

## Data Availability

Data is available from the author.

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
