# Peer review of "Integration of Mesoporous Bioactive Glass Nanoparticles and Curcumin into PHBV Microspheres as Biocompatible Composite for Drug Delivery Applications"

_molecules, 2021, doi:10.3390/molecules26113177_

Round 1

Reviewer 1 Report

Comments to Authors on the Manuscript Number: 1206520

The paper “Integration of Mesoporous Bioactive Glass Nanoparticles and Curcumin into PHBV Microspheres as Biocompatible Composite for Drug Delivery Applications”, by A.E. Aguilar-Rabiela, A. Leal-Egana, Q. Nawaz and A.R. Boccaccini, approaches the synthesis and preliminary characterization of a ternary system composed of glass nanoparticles, polyhydroxyalkanoate microspheres and curcumin molecules. Besides the compositional and morphological evaluation, the final samples were assessed from the biocompatibility, bioactivity and drug release point of view.

The article presents major weaknesses, as follows:

  1. The first two phrases from the Abstract section should be removed.
  2. The Introduction section is quite limited and does not provide a satisfactory background for the addressed topic. I suggest rewriting this section in its entirety, by including advantages/disadvantages, significant case studies, targeted objectives based on each component, as well as highlighting the work novelty.
  3. Please explain the selection of those particular ratios between PHBV and MBGN; another way of expressing these reports, much easier to understand, would be desirable.
  4. Why only the sample with a ratio of 10:1 was assessed from the bioactivity point of view? The PHBV:MBGN ratio influence on the mineralization rate would have been an interesting finding.
  5. Which composite was tested in Fig. 1? All PHBV:MBGN ratios led to the same average diameter?
  6. Why only 3 compositions were included in Fig. 2? Moreover, the scale bar is different from the first to the next images, fact that makes the comparison difficult. The quality of image b) is questionable.
  7. XRD pattern does not provide information about functional groups (“Due the fact that no new functional groups in the FTIR spectrum and XRD diffractogram appeared, it is possible to consider the interaction of the composite as a blend of the different components without the formation of any other new phase.”). The phrase must be corrected.
  8. Which is the meaning of the colours employed in Fig. 4? I suggest positioning the Miller indices above the peaks.
  9. Again, which is the investigated composite in Figs. 3 and 4?
  10. The graph from Fig. 6 is not explained in any way in the associated paragraph.
  11. Why only 3 compositions were integrated in Figs. 7 and 9?
  12. The caption of Fig. 8 must be revised.
  13. Please explain the meaning of the colours visible in Figs. 8 and 10.
  14. How do you correlate the results of cell viability with those provided by the fluorescence microscopy? The values of cell viability are relatively similar from 24 h to 7 days, while the fluorescent images show two different situations.
  15. In Fig. 11, the XRD pattern must be indexed, while the SEM image should be presented at a higher magnification to identify the nanometric components of the globular structures. How do you explain this quite fast mineralization, when you claim that “The PHBV matrix serves as an interface which moderates the release and reduces the direct contact of MBGN with the cell culture medium.”?
  16. The axis titles must be corrected in several figures to ensure consistency.
  17. Overall, the paper seems superficially written, even though the approached subject is interesting and presents potential for future development. Many of the explanations are theoretical knowledge or simple observations from figures; the interpretation and comparison part needs to be seriously improved, in addition to clarifying the exclusion of some compositions from certain discussions.

In conclusion, the paper “Integration of Mesoporous Bioactive Glass Nanoparticles and Curcumin into PHBV Microspheres as Biocompatible Composite for drug Delivery Applications”, by A.E. Aguilar-Rabiela, A. Leal-Egana, Q. Nawaz and A.R. Boccaccini, can be published in Molecules (MDPI) after a major revision.

Author Response

Reviewer 1 Reply

1.- The first sentences of the abstract have been modified.

2.- The introduction section was rewritten in other to clarify the advantages and disadvantages of each material by separate, the purpose of each material involved in the composite, relevant state of the art about similar composites or previous approaches were added and the difference of this work compared with previous works was remarked. 

3.- The proportion of MGBN into the PHBV microspheres was selected in the way that the amount of MBGN in the microspheres would increase from 10% to 50% (w) to obtain five conditions (10, 20, 30, 40, 50) with steps of 10% (w) of MBGN increasing between each other. The explanation about this has been included in the materials and method section and the nomenclature was modified to (%w/%w) of PHBV/MBGN to be easier to understand.

4.- The samples with the low proportion of MBGN (90:10) were selected to find evidence of bioactivity. This was the condition with the lowest presence of MBGN, as bioactivity was observed in this condition, it was assumed that for higher proportions of MBGN, bioactivity would be also present. This assumption is mentioned in the results and discussion section.

5.- The average diameter was obtained from the five conditions studied, and there was no significant change in diameter. This has been clarified in the results section and a micrograph of blank PHBV microspheres was added by suggestion of the other reviewer, also supported with extra references.

6.- The discussion about the morphology of the different conditions of the composite microspheres was extended to clarify which conditions exhibited similar size and shape and what others exhibited a different tendency. An additional micrograph was included in figure 2 in order to compare the morphology at the same scale, and different conditions at their respective scale.

7.- The section about FTIR and XRD was corrected including comparison to previous works.

8.- Red graph was the softness of the pattern, colors have been modified and Miller Indices positioned above each peak.

9.- The samples used for the experiment were clarified in the results section.

10.- Figure 6 is included in the results section but the release behavior is discussed in the discussion section.

11.- Figure 7 and Figure 9 were initially considered to show the conditions with significant differences between the positive and negative controls (Control, free MBGN) (50:50, 80:20, 90:10). However, the full graph with all conditions was added now in both Figure 7 and 9.

12.- Caption of Figure 8 was extended.

13.- Staining colors of figures 8 and 10 were explained in the results section.

14.- The WST-8 cell viability test of free MBGN showed a reduction on the cell viability of up to 30% compared to the 90:10 condition and up to 10% compared to the 50:50 condition during the first 24 h. However, after 7 days a reduction of the viability on the cells treated with free MBGN was up to 45% compared to the 90:10 condition and about 20% compared to the 50:50 condition. We considered that these results are not similar. On the other hand, the fluorescent micrographs of specific samples were shown in order to compare the morphology of the cells exposed to free MBGN and those exposed to composite microspheres. The differences in the morphology and the population brings evidence of the different cell interactions with MBGN and microspheres. This has been now extended in the discussion section.

15.- Figure 11 was modified and the XRD pattern was adequately indexed. The morphology of the layer of hydroxyapatite (HA) is similar to that shown in previous reports after similar time lapses (see ref. [1]), however it was not possible to establish through SEM micrographs the thickness of the layers. In the best case the layer indicates an initial stage of the mineralization. This also corresponded with the XRD pattern in which the intensity of HA is not that high compared to those reported in previous studies. This appears to indicate a larger influence due to the differences of size and the resulting interaction with cells, than the effect of release of ions. However, a more detailed discussion about this aspect is not possible with the results presented in this work.

16.- The axes of all figures were revised.

17.- The draft has been improved in general, additional discussion was included, and additional literature referred to compare the results with previous works. We hope all these modifications have improved the article and it can be accepted for publication.

  1. Nawaz, Q., et al., Synthesis and characterization of manganese containing mesoporous bioactive glass nanoparticles for biomedical applications. Journal of Materials Science: Materials in Medicine, 2018. 29(5): p. 64.

Reviewer 2 Report

The paper submitted by Aquilar-Rabiela et al. investigates the preparation of some composite microparticles based on PHBV and loaded with bioactive glass nanoparticles (MBGN) and curcumin, as a model drug.

The manuscript is clear, well written, and the conclusions are supported by the results. However, some corrections are needed in order to increase the overall quality before publication:

  1. Both the introduction and discussion section can be improved by citing different references concerning the loading of curcumin in various nano and microcarriers. Some examples can be: https://doi.org/10.3390/polym12071450; https://doi.org/10.1016/j.ijbiomac.2019.12.247; https://doi.org/10.3390/ijms22063075
  2. The authors must provide the size and the ZP value of the blank PHBV microspheres in order to discuss the influence of both MBGN and curcumin on this parameter.
  3. Line 72: the values of the ratios given at this point are not the same with the values provided in the caption of fig 2. Please revise. The SEM micrographs of blank microspheres should also be added.
  4. In fig 3, the spectra of blank microspheres, curcumin and MBGN must be given and the possible shift of the characteristic peaks of these components in the spectra of the composite microspheres should be discussed in order to determine if there are some interactions between the components.
  5. The caption of the fig 6 must be completed by adding the PHBV/MBGN ratio. It should be also of interest if the release curve for the lowest PHBV/MBGN ratio can be given.
  6. Please add in the materials section the Mw and DH of the PVA.
  7. Line 309: the ratio 10:1 used for curcumin is compared to PHBV?! It’s not clear from this sentence…

Author Response

Reviewer 2 Reply

1.- Some suggested references, related to entrapping of curcumin in micro and nano devices were included in the new version.

2.- The Z potential values of MBGN and PHBV microspheres with curcumin were added and also the influence on the composite was discussed in the respective section and compared with previous works. The size of blank PHBV capsules was added to the results section however, due to the fact that the size of microspheres fabricated through O/W emulsion is more influenced by the process (i.e., stirring, emulsifier) [1, 2], and no difference in size of all conditions was observed, as much in our experiments as in previous reports (also referred in the draft) [3-6], the influence of the loading on the microsphere size was not further discussed.

3.- The result section (in line 72) was corrected according to what Figure 2 describes. A micrograph of a blank PHBV microsphere was added and briefly mentioned due to the fact that no difference was observed between blank PHBV and composite microspheres at the discussed ratios.

4.- The characteristics excitation wavelengths for each component have been previously described in different studies involving these materials and no formation of new functional groups has been reported. Due this, the purpose of the FTIR analysis in this study is, in combination with the curcumin kinetic release and the bioactivity assessment, to support the presence of the components in the composite. Besides, there are different examples in literature in which the interaction of the components has been described as a blend [3, 7], many of these references are cited in the new version of the paper.

5.- The caption of Figure 6 was extended.

6.- The PHBV/curcumin ratio used in this study was clarified in the materials and methods section as in the curcumin entrapment efficiency section.

  1. O'Donnell, P.B. and J.W. McGinity, Preparation of microspheres by the solvent evaporation technique. Advanced Drug Delivery Reviews, 1997. 28(1): p. 25-42.
  2. Francis, L., et al., Controlled Delivery of Gentamicin Using Poly(3-hydroxybutyrate) Microspheres. International Journal of Molecular Sciences, 2011. 12(7): p. 4294-4314.
  3. Macías‐Andrés, V.I., et al., Preparation and characterization of 45S5 bioactive glass‐based scaffolds loaded with PHBV microspheres with daidzein release function. Journal of Biomedical Materials Research Part A, 2017. 105(6): p. 1765-1774.
  4. Lee, Y. and H. Sah, Simple emulsion technique as an innovative template for preparation of porous, spongelike poly(lactide-co-glycolide) microspheres with pore-closing capability. Journal of Materials Science, 2016. 51(13): p. 6257-6274.
  5. Swornakumari, C., et al., Preparation of microspheres using poly-3-hydroxybutyrate biopolymer and its characterization. Journal of Environmental Biology, 2018. 39(3): p. 331-338.
  6. Aguilar-Rabiela, A.E., et al., Modeling the release of curcumin from microparticles of poly(hydroxybutyrate) [PHB]. International Journal of Biological Macromolecules, 2020. 144: p. 47-52.
  7. Li, W., et al., Preparation and characterization of PHBV microsphere/45S5 bioactive glass composite scaffolds with vancomycin releasing function. Materials Science and Engineering: C, 2014. 41: p. 320-328.

Round 2

Reviewer 1 Report

Personally, I am not satisfied with the feedback provided on several comments: 1 (theory in the abstract), 2 (introduction too little extended), 5 (Fig. 1 is still confusing, since it is not clear if the curve is an average one or other thing), 6 (the caption of Fig. 2 presents errors, while the addition of a supplementary image does not improve the situation in the way it was suggested), 9 (the situation must be clarified locally, in the figures caption), 14 (the comment intended to draw attention to a contradiction between the results generated by the two methods) and 15 (there are no Miller indices in Fig. 11, while the quality of the SEM image is poor).

I consider that the paper will be suitable for publication only after a second revision.

Author Response

Second revision

1.- The abstract was written with the intention of being short and to provide the key results of the study. Theoretical concepts do not usually belong in the abstract but are included in the Introduction section.

2.- The introduction section was extended to include the background information we considered to be sufficient for the readers to follow the rest of the paper. The paper includes a brief but clear introduction about the main issues involved in this experimental work. We tried not to repeat conventional information about polymer-BG composites for drug delivery as this has been presented in many previous papers, some which are included as references in this paper. We do not think that expansion of the Introduction will increase understanding of the scope of the paper.

5.- Caption of Figure 1 was modified to clarify the meaning of the curve presented.

6.- Caption of Figure 2 was modified according to the comparison discussed in the respective section. The purpose of Figure 2a and Figure 2b is to show the spherical shape and semi-homogeneity and shape of composite microspheres at the PHBV/MBGN ratios used in this work and compared them to blank PHBV microspheres. The purpose of Figure 2c and Figure 2d is to show the special cases of no homogeneity exhibited in PHBV/MBGN composite microspheres at the highest ratios studied in this work. This is also discussed now in the result section and in the discussion section. We do not see what else can be added to the caption?

9.- Captions to Figures 3 and 4 were modified to clarify locally which ratio was used for the analysis.

14.- It is no possible to correlate the results of an enzymatic cell viability method (WST-8), calculated by the optical density average, with fluorescent microscopy images resulted from a DNA staining. WST-8 is a quantitative analysis and fluorescent micrographs observations are presented as qualitative results in this work. The main discussion is supported by the WST-8 results and the contribution of the fluorescent micrographs is to bring additional evidence about the proliferation and morphology of the cells after the exposure to high concentrations of MBGN compared to same amounts of MBGN trough the composite, which has not been studied in previous works. A quantitative analysis from a batch of micrographs with additional processing could be explored in future studies. This was explained in de discussion section.

15.- Figure 11 was modified and SEM micrograph improved in quality, miller indices were added to the figure.

We hope that these modifications are sufficient and that this version of the paper can be accepted for publication.  
